# A Software-Defined Directional Q-Learning Grid-Based Routing Platform and Its Two-Hop Trajectory-Based Routing Algorithm for Vehicular Ad Hoc Networks

**DOI:** 10.3390/s22218222

**Published:** 2022-10-27

**Authors:** Chen-Pin Yang, Chin-En Yen, Ing-Chau Chang

**Affiliations:** 1Department of Computer Science and Information Engineering, National Changhua University of Education, Changhua 50007, Taiwan; 2Department of Early Childhood Development and Education, Chaoyang University of Technology, Taichung 41349, Taiwan

**Keywords:** vehicle to everything (V2X), reinforcement learning, Q-learning, Software-Defined Network (SDN), Software-Defined Directional QGrid (SD-QGrid), source–destination directionality, two-hop trajectory-based routing (THTR)

## Abstract

Dealing with the packet-routing problem is challenging in the V2X (Vehicle-to-Everything) network environment, where it suffers from the high mobility of vehicles and varied vehicle density at different times. Many related studies have been proposed to apply artificial intelligence models, such as Q-learning, which is a well-known reinforcement learning model, to analyze the historical trajectory data of vehicles and to further design an efficient packet-routing algorithm for V2X. In order to reduce the number of Q-tables generated by Q-learning, grid-based routing algorithms such as the QGrid have been proposed accordingly to divide the entire network environment into equal grids. This paper focuses on improving the defects of these grid-based routing algorithms, which only consider the vehicle density of each grid in Q-learning. Hence, we propose a Software-Defined Directional QGrid (SD-QGrid) routing platform in this paper. By deploying an SDN Control Node (CN) to perform centralized control for V2X, the SD-QGrid considers the directionality from the source to the destination, real-time positions and historical trajectory records between the adjacent grids of all vehicles. The SD-QGrid further proposes the flows of the offline Q-learning training process and the online routing decision process. The two-hop trajectory-based routing (THTR) algorithm, which depends on the source–destination directionality and the movement direction of the vehicle for the next two grids, is proposed as a vehicle node to forward its packets to the best next-hop neighbor node in real time. Finally, we use the real vehicle trajectory data of Taipei City to conduct extensive simulation experiments with respect to four transmission parameters. The simulation results prove that the SD-QGrid achieved an over 10% improvement in the average packet delivery ratio and an over 25% reduction in the average end-to-end delay at the cost of less than 2% in average overhead, compared with two well-known Q-learning grid-based routing algorithms.

## 1. Introduction

The new generation of the 5G network allows for a higher performance and wider communication range, but how to maintain service quality in the V2X (Vehicle-to-Everything) [1] environment whilst considering the concept of differentiated service orientation and the establishment of package routing is still an urgent problem [2]. The Internet of Vehicles is an ad hoc network established by vehicles as packet transmission nodes. It has a dynamic topology. In the network environment, vehicle nodes execute routing algorithms individually. In addition, the nodes in the Internet of Vehicles move fast and are unstable [3,4]; therefore, the packet transmission paths of the ad hoc network are not fixed. Hence, it is very difficult to maintain the routing table; in particular, the routing protocol, which needs to be updated frequently, is more difficult to apply in this type of environment.

In order to solve the connection problem [5], this problem was proposed to be solved using the Delay-Tolerant Network (DTN) [6] in past related research. The Internet of Vehicles benefits from the thinking method of SCF (Store-Carry-and -Forward) [7], which is a network that does not need to establish a connection path in the ad hoc network between the source and the destination to transmit information. The information is transmitted by the contact opportunity between nodes. When there is no neighbor node or the neighbor node is in a poor condition, it will keep the packet and wait for the next opportunity to contact other neighbor nodes and transmit the packet. The most basic DTN method is flooding. Later, the improved DTN method is used to compare the value of a certain cost between the current vehicle and the adjacent vehicle, or to calculate the probability of the meeting of nodes, and is also used to decide the adjacent vehicle to receive the transmitted packet. There is no evaluation or improvement strategy for the packet delivery path. Overall, the trade-offs in the DTN can be divided into: (i) storage resources vs. the packet transmission success rate; and (ii) transmission delay vs. the delivery probability, which requires further consideration.

Reinforcement learning [8] is a research direction related to machine learning, which has a great impact on artificial intelligence (AI) and is applied to improve the routing algorithm for different ad hoc network architectures, such as wireless sensor networks [9], VANET [10], flying ad hoc networks [11], and drone ad hoc networks [12]. The survey [13] compiled several algorithms that use Q-Learning as a routing decision element. The ADOPEL [14] algorithm assumes that there is a TCC (traffic control center) in the network environment, and vehicle nodes transmit traffic data to the TCC. The TCC has a macroscopic view of the whole network, and uses the delay of the links and the number of aggregable data packets as the Q-learning parameters. After generating the Q-table, the vehicle nodes will select the next-hop nodes accordingly. The QTAR [15] algorithm applies Q-learning to vehicle-to-vehicle (V2V) and RSU-to-RSU (R2R). Considering link quality, expiration time and delay, it also adds the concept of SCF to the packet delivery of vehicle nodes. The QGrid [16] routing algorithm divides the map into many grids, and uses the number of vehicles in the grid as a parameter to generate the Q-table through Q-learning [17], determining the delivery direction between the four neighboring packet areas.

The Software-Defined Network (SDN) is a new network architecture [18,19,20,21], which is generally regarded as the future network type in the industry. The method replaces the traditional network equipment performing individual link discovery and routing calculations, with the SDN controller in charge of performing these calculations. The architecture, the central control mode, uses programs to reschedule the network, including for packet-routing and routing-table maintenance. It addresses the shortcomings of the traditional centralized routing algorithm, and can also cope with the large amount of data exchange in the server. At present, VANET mainly executes the routing algorithm via each vehicle node, and establishes the decision of the next-hop neighbor node through the exchange of messages between nodes. Although the load of a single routing host is dispersed, under the distributed routing architecture, several nodes may possibly select a specific vehicle node to forward packets to at the same time because they lack knowledge on the result of each other’s routings. This may easily cause congestion and the unsatisfactory arrangement of the packet delivery route.

### 1.1. Problem Background and Motivations

In a network environment dominated by vehicles with relatively fixed trajectories, such as buses, how to use trajectory information to make routing decisions is a problem that needs to be considered. According to our observations, the shortcomings of the above related research are: (i) Intergrid routing does not consider the overall direction of the vehicles’ movement between grids to select the best neighbor grid, and does not use the corresponding Q-tables for different time periods. (ii) Intragrid routing does not focus on how to analyze the vehicles’ actual trajectories for selecting the best neighbor node in grids to forward packets to. Therefore, grid-based packet transmission needs to refer to the following three directionalities:

Grid directionality and source-to-destination directionality for intergrid routing: By analyzing vehicle trajectories in the overall network environment, we define the number of outgoing vehicles from the current grid to the adjacent grid as the *grid directionality* of this adjacent grid in this paper. We believe that the grid directionality has an important impact on increasing the successful transmission rate and reducing the delay of the packet delivery. If you select a neighbor grid that is more in line with the vehicle’s driving direction, i.e., the grid with higher grid directionality, as the next-hop grid to forward the packet to, the packet has a higher probability of reaching the destination. In addition, the direction from the source node to the destination node, i.e., the source-to-destination directionality in this paper, also guides the best selection of the next-hop neighbor grid. If Q-learning selects a neighbor grid that has the higher grid and the source-to-destination directionalities, this grid will obtain a higher reward and Q-value, which are used as the decision basis of the intergrid routing. As shown in Figure 1, the starting position of the packet is at $Grid_{D}$, and it will be sent to the destination at $Grid_{C}$. Because the Q-value of $Grid_{E}$ is the highest one among all neighbor grids of $Grid_{D}$, the intergrid routing for $Grid_{D}$ selects $Grid_{E}$ as the best next-hop grid. This means that the intergrid routing can use the grid directionality to train the Q-model and select the most plausible neighbor grid to forward the packets to.Two-hop directionality for intragrid routing: By analyzing the historical vehicle trajectories to determine the grid directionality of each grid, we can select the neighbor vehicle whose future driving trajectory goes to the best next-hop grid selected by the intergrid routing. However, there may be more than one possible neighbor vehicle that meets the above conditions; thus, how to make an appropriate choice is the important issue to address. As shown in Figure 2, vehicle node $v_{a}$ located in $Grid_{A}$ has two neighbor nodes, $v_{b}$ and $v_{c}$, located in the best next-hop grid $Grid_{C}$. Which of the two is the best relay node? If a neighbor node whose future driving trajectory moves to the best next-hop grid first, and then, to the best neighbor grid next, which is called the best two-hop next grid in this article, this node will be a better relay node than any node whose trajectory does not follow this two-hop path. This is because this relay node would have a higher probability to meet the destination node than others, even though it does not meet any neighbor node on this two-hop path to forward the packet to, but has to carry the packet by itself to the destination using SCF. Hence, this paper proposes two-hop trajectory-based routing (THTR) as follows. THTR first performs the intergrid routing on the best next-hop grid, i.e., $Grid_{C}$, of $Grid_{A}$ to select the next best neighbor grid, i.e., $Grid_{E}$. It then analyzes the historical trajectories of $v_{b}$ and $v_{c}$ located in the best next-hop grid, $Grid_{C}$, to find their future positions. If $v_{c}$ will enter the best two-hop next grid $Grid_{E}$ at position $v_{c^{\prime }}$, it would be a better strategy for $v_{a}$ to forward the packet to node $v_{c}$ instead of $v_{b}$, because the future position of $v_{b}$, that is, $v_{b^{\prime }}$, enters $Grid_{B}$ later.

**Figure 1 sensors-22-08222-f001:**
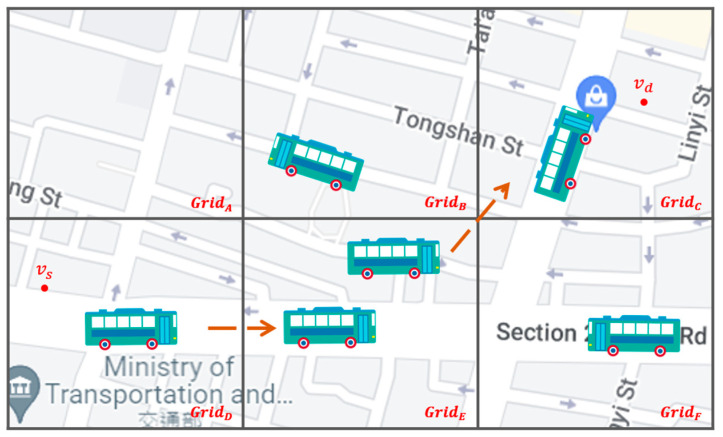
Grid directionality and source-to-destination directionality for intergrid routing of SD-QGrid.

**Figure 2 sensors-22-08222-f002:**
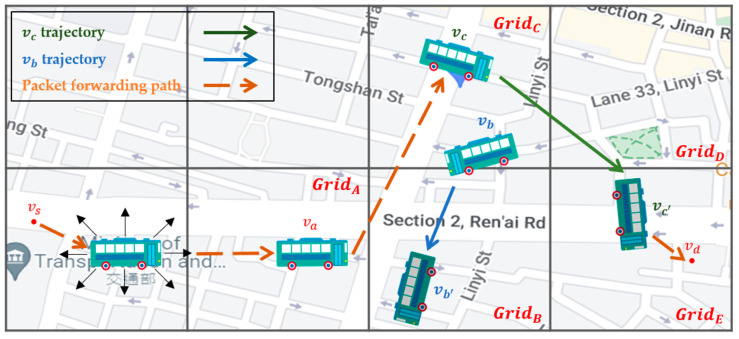
The two-hop directionality for the intragrid routing of SD-QGrid.

The routing algorithm proposed in the original QGrid paper only uses the number of all nodes in each grid to represent the probability of the packet reaching the destination if the grid holds a packet. However, we think that simply considering the number of vehicle nodes in the grid is too one-sided. Further, it is found from the road distribution of the map that not all roads are horizontal or vertical, and some adjacent grids are connected by diagonal roads. The intergrid routing of QGrid only uses four neighbor grids to select the next-hop grid, ignoring the possibility of transmitting packets to the neighbor grids in the diagonal direction, which will lead to a decrease in the transmission success rate and an increase in delay. In addition to taking the number of vehicle nodes in the grid as the basis for decision making, we also refer to the vehicle mobility, i.e., the grid directionality, between the current grid and eight neighbor grids to avoid ignoring the vehicle nodes located in the diagonal grid, which happens with the QGrid that only considers neighbor grids in four directions. Therefore, the improved method we proposed through the SD-QGrid is to further consider the average directionality of each grid, based on the number of outgoing nodes, and expand the neighbor grid to eight directions, as shown in Figure 3, in the Q-table calculation. Compared with the original Q-learning in the four directions of up, down, left, and right, we added a diagonal pair. There are eight directions of angular movement (such as the red arrow), and the performance of the Q-table in the SD-QGrid intergrid routing is better.

### 1.2. Contributions

To design our network structure, the concept of SDN routing architecture is added. We hope to consider the historical density and directionality of vehicles between grids, and modify the way that Q-learning generates the Q-table to determine the macrotransfer direction of the V2X packet. Then, according to the historical trajectories of the real vehicles, we can select the most appropriate packet delivery path, that is, the microscopic delivery path. This paper makes the following contributions:We design the Software-Defined Directional QGrid (SD-QGrid) network architecture, which combines the centralized control concept of the SDN Control Node (CN), and proposes the offline Q-learning training process and the online routing decision process in a V2X network architecture with reference to vehicle trajectories and three types of directionalities.Using the number of vehicles and the trajectory and directionality of vehicles between grids as parameters, the offline Q-learning training process of this paper modifies the Q-learning of reinforcement learning by increasing the moving directions between neighboring grids from the original four to eight and using the corresponding Q-tables for different time periods [22] as the macroreference for intergrid routing packet delivery.In the online routing decision process, a two-hop trajectory-based routing (THTR) algorithm is proposed, referring to the historical trajectory of the vehicle and the two-hop transmission path for calculating the future trajectory. This algorithm selects the neighbor node which is most suitable for the direction along the best next-hop grid and the best two-hop next grid, and then forwards packets to it, increasing the probability to meet the destination node and resulting in a shorter packet transmission path and lower packet transmission delay.We conduct simulation experiments to analyze and find the routing paths in V2X, and then obtain real and meaningful performance data of the routing algorithm using the real vehicle movement trajectories in Taipei City.

## 2. Related Work

Regarding the algorithm of the vehicle communication routing decision, we can introduce the characteristics of the vehicle trajectory and design our algorithm accordingly. Since humans have a habit of moving, there is a very high probability that they will follow a similar route to the same place. Since vehicles extend humans’ social behavior, vehicle travel paths are largely predictable. Vehicles can be divided into three categories according to the stability of their trajectories (a trajectory is a series of locations where the vehicle passes):Buses: the trajectory is very accurate, with a relatively fixed arrival time and travel path.General private cars: have regular trajectories and obvious regularity in time and space.Taxis: have a flexible and variable travel route, and often have a credible destination [23], but the travel path is not fixed.

### 2.1. Position-Based Routing Algorithms without Adopting Reinforcement Learning

(1) DPPR [23] assumes that each vehicle knows its own driving trajectory in advance, and the driving trajectory is a sequence formed by the number of intersections that will be passed through. Due to the traffic information, each road can be assigned a weight value, which is the transmission delay time of this road, and the shortest delay path can be calculated via the Dijkstra algorithm [24]. DPPR contains two different modes: the straightway mode and intersection mode. When the vehicle wants to forward the packet, it first calculates the shortest delay path to determine the next expected intersection arrival time, and detects whether its neighbor vehicles will pass the destination of the packet. If so, the vehicle carrying the packet will forward the packet to this vehicle; if not, the neighbor vehicle that is driving toward the target intersection and is faster than the vehicle carrying the packet is selected, and the packet is copied to the neighbor vehicle; if both vehicles cannot be detected, the vehicle carrying the packet will place the packet into its routing queue and schedule. The disadvantage of this method is that it needs to know the complete driving trajectory, and then calculate the shortest delay path, which is not suitable for vehicles without fixed routes.

(2) The NTR (Novel Trajectory Routing) [22] algorithm uses the vehicle GPS trajectory to explore all the possible movement patterns of each vehicle, establish its own trajectory tree, and establish a vehicle encounter tree by comparing the trajectory trees of the vehicles. Compared with the vehicle trajectory representation used in the previous paper, only the location where the vehicle passed by is recorded, and the NTR also records the elapsed time in the trajectory. The vehicle encounter tree records all possible paths in which the packet is forwarded from the source to the destination, and records the road segment between the vehicle carrying the packet and the expected vehicle to be forwarded the packet. After building the vehicle encounter tree, NTR converts the vehicle encounter tree into a predicted transit vehicle sequence diagram. The packet sender deletes the key of the vehicle child nodes that will meet along the road; changes them into the parent node of the vehicle that the packet sender will meet directly; and adds two parameters, which are the expected packet arrival rate (expected delivery ratio, EDR) and token (token value indicates the number of packet replicas). If the expected packet arrival rate of the destination node is 1, the expected packet arrival rate of each node toward the root node in the direction of the parent node is continuously calculated. When the expected packet-forwarding sequence diagram and related information are calculated, the packet transmitting end can use the sequence diagram and related information to transfer the packet. The advantage of the NTR algorithm is that it can calculate the transfer vehicle sequence between any two vehicles and achieve a transfer process close to the optimal solution. The disadvantage of this is that the calculations of establishing a trajectory tree, comparing the trajectory tree of vehicles, and establishing a vehicle encounter tree are complicated. They require a large amount of computing time and storage space for the pre-operation.

### 2.2. Position-Based Routing Algorithms with Reinforcement Learning

In the following, we introduce the concept of reinforcement learning, Q-learning, and then compare several position-based routing algorithms that use Q-learning as the basis for decision making.

First, Q-learning is briefly described. The reinforcement learning model is composed of several elements. We will first define a few terms below.

Definition: a state is called a Markov state if and only if:
$$P[S_{t+1}|S_{t}]=P\left[S_{t+1}|S_{1},\;S_{2},\;S_{3},\;\ldots ,\;S_{t}\right]$$

Definition: For a Markov state s and its successor state s’, define the State Transition Probability as $P_{SS^{’}}=P[S_{t+1}=S’|S_{t}=S]$. The State Transition Matrix P is the transition probability for all states s to all successor states s’.

Definition: a policy π is the probability distribution of action a after a given state s.
$$\pi \left(s\right)=P\left[A_{t}=a|S_{t}=s\right]$$

Definition: the reward $G_{t}$ is the sum of accumulated depreciation rewards starting at time t.
$$G_{t}=R_{t+1}+rR_{t+2}+\cdots ,\;where\;r\in \left[0,1\right]$$

Definition: the action–value function $q_{\pi }\left(s,\;a\right)$ is the expected reward obtained by starting from state s, selecting action a, and then following policy π.
$$q_{\pi }\left(s,\;a\right)=E_{\pi }\;\left[S_{t}=s,\;A_{t}=a\right]$$

According to the Bellman equation, we can finally rewrite the above formula as:(1)$$q_{\pi }\left(s,\;a\right)=R_{s}^{a}+r\underset{s^{\prime }\in S}{\sum }P_{Ss^{’}}^{a}\;\underset{a^{\prime }\in A\;}{\sum }\pi \left(s’\right)q_{\pi }\left(s’,\;a’\right)$$

Finally, we will discuss Q-learning training, for which the core update formula is:(2)$$Q\left(s_{t},\;a_{t}\right)\leftarrow \left(1-\alpha \right)*Q\left(s_{t},\;a_{t}\right)+\alpha *\left(R_{t+1}+\gamma *max_{a}Q\left(s_{t+1},\;a’\right)-Q\left(s_{t},\;a_{t}\right)\right)$$

α is the learning rate (0 < α < 1). γ is the discount factor (0 < γ < 1). When the value of γ is larger, the long-term rewards obtained in the future will receive more attention; the smaller the value, the more the current rewards will be considered. R is the reward; it acts on each state and will receive the corresponding reward.

Position-based routing algorithms use Q-learning as the basis for decision making:
(1)The ITAR-FQ [25] algorithm architecture consists of two main parts: the real-time traffic aware process and road evaluation (RTAP-RE) and the routing decision process with fuzzy Q-learning (RDP-FQ). The RTAP-RE designs a road evaluation method to process traffic information and estimate road quality (RQ), with reference to the number of vehicles moving in the same/a different direction, the length of the road, packet generation time, current time, etc. RDP-FQ divides the entire routing process into multiple routing processes. As the intersection is divided into multiple routing processes, the packet continues to look for the next intersection and selects a new road segment until the packet reaches the destination. Facing the intersection, it calculates the road score (RC) by combining RQ and the Manhattan distance (${\mathrm{MD}}_{\mathrm{normal}}$), and selects the road with the highest RC as the next forward road segment. The reward function is modified by referring to the benchmark reward and the corresponding value given by Fuzzy Logic. The advantage of this paper is that the Fuzzy Logic indicator is added to Q-learning for routing decisions in road segments. The disadvantage is that the directionality of the relative position of the transmitter and the destination are not considered. If the road segment with the destination end is the next hop at the intersection, it may cause the packet to be transmitted over a long distance or repeated on the road segment, and the historical movement trajectory of the vehicle is not used for training to obtain better routing decisions.(2)ADOPEL [14] assumes that two kinds of packets are exchanged between vehicle nodes: beacons and event-driven messages. The former is for the exchange of information such as the position, speed, and direction of travel between vehicles. The latter is used for vehicle nodes to collect traffic data and transmit it to the traffic control center (TCC), so that the TCC can gain a macroscopic understanding of the entire vehicle network environment, analyze the current number of neighbor vehicle nodes and the transmission delay of each node, and substitute this decision data into Q-learning, before using the Q-value to select better next-hop neighbor nodes for each vehicle node. The advantage of this approach is that it can better adapt to the high mobility and topology variability of the VANET network environment, whereas the disadvantage is that it does not refer to the future trajectory of the vehicle.(3)In the QTAR [15] network architecture, Q-learning is used for vehicle-to-vehicle (V2V) and RSU-to-RSU (R2R) networks. Decision elements, such as connection reliability and EED, are added to the Q-learning operation, and then, next-hop neighbor nodes are selected and packets are forwarded. The advantage of this approach is that the Q-learning planning method can improve throughput and PDR, but the disadvantage is that the directionality of the nodes is not considered, resulting in an unsatisfactory overall transmission performance when used in a real-world environment.(4)The QGrid [16] divides the entire map environment into equal grids, and uses Q-learning in advance. It uses the number of vehicles in the area as a parameter, and calculates the Q-table in advance to determine the transfer direction between grids, which is a macrotransfer consideration. QGrid_G uses the greedy method of packet transmission, which preferentially selects the neighbor node closest to the destination. QGrid_M uses the two-hop Markov prediction [26,27,28] method to predict the grid that the packet will pass through in the future, and preferentially selects the neighbor node with a higher probability of entering the next-hop grid as the next-hop node. The above two methods are microscopic transfer considerations. The advantage of these is that the transmission of packets is determined by considering the different macro- and microlevel aspects at the same time. Considering the number of vehicle nodes in each grid, the packets are preferentially transmitted to the grid with more vehicle nodes so that the packet loss rate is reduced. The disadvantage is that the vehicle density is only considered, and the influence of the overall movement direction between grids on the overall environment is not considered. The advanced QGrid is a routing protocol improvement made by QGrid, which is for vehicles with relatively regular and fixed trajectories, such as bus nodes, in the network environment. The vehicle node carries the packet to the destination grid and then continues the next-hop packet forwarding. The advantage of this is that it considers the reference vehicle trajectory, as well as the next-hop node selection, based on the trajectory information. The disadvantage is that even if the vehicle node has neighbor nodes, it still needs to continue to carry the packet until it reaches the grid destination where the node is located, resulting in a longer packet transmission delay time.

We have discussed several routing algorithms that analyze vehicle trajectories and formulate designs based on them. Table 1 is a comparison of related studies. It can be found that most of the V2X routing methods lack the analysis of real vehicle trajectory data, even if they use reinforcement learning. They also lack the analysis of the source–destination direction, as well as a software-defined routing platform and two-hop next-grid routing.

## 3. SD-QGrid Routing Platform and Algorithms

### 3.1. SD-QGrid Network Architecture

First, we define the grid in the SD-QGrid network architecture. As shown in Figure 4, we calculate the length and width of the map range of this experiment; we know the latitude and longitude of the four vertices of the map; we take the vertex P in the upper left corner as the reference point; we calculate the distance between the point P and the vertex in the upper right corner to obtain the length L of the outgoing map; and we obtain the width W of the map by calculating the distance between the point P and the vertex in the lower left corner. After determining the length L and width W of the experimental map, we can then give the grid length d to mark m × n grids in this map. The grid number range is (0 … m − 1, 0 … n − 1).

The entire SD-QGrid has an SDN Control Node (CN), which is responsible for initializing the grid structure of the entire map by defining the grid size, cutting the experimental map, and calculating the grid where all vehicle nodes are located. In addition, each grid needs to be equipped with an RSU, which can communicate with the CN through wired lines, at each intersection to collect vehicle trajectory information of all vehicles located at road segments connected to this intersection. Each vehicle on a road segment assumes to own the communication module and the GPS receiver such that it can periodically broadcast its instant information, including the vehicle ID, velocity, GPS position, moving direction, etc., to the first reached RSU through the HELLO messages. In this way, the loss of vehicle information caused by building obstacles in the grid can be reduced [29].

The SD-QGrid is proposed to contain two operation processes. The first one is the offline Q-learning training process and the second one is the online routing decision process. Details of these two processes are explained below.

The offline Q-learning training process: Using the vehicle trajectory and grid-related information, the offline Q-learning training process is responsible for training the Q-learning model to generate Q-tables of each grid at the SDN CN when the whole SD-QGrid system initializes or a new vehicle enters this V2X environment. The SD-QGrid may periodically perform this process to update Q-tables.The online routing decision process: Whenever a vehicle node carrying a packet intends to transmit this packet to the destination node, the SD-QGrid executes the online routing decision process to select the best neighbor node as the relay. Consequently, the packet will efficiently reach its destination at the end of the routing process.

### 3.2. Experimental Map

This project uses the data of Taipei City’s bus movement trajectory [30]. First, we planned a latitude and longitude range in Taipei City, and the vehicles appearing in this range were used as research data and recorded in tables. The first table records the name of the route, the license plate number, the direction of departure and return, and the time when it first appeared in the data, which is used to record the departure sequence of the license plate number corresponding to the bus route, and records the return time in the direction to and from this vehicle node. The other table records the longitude and latitude of each vehicle node at each time point, and the coordinates of the grid where the vehicle node is located, which is used to record the actual trajectory of each vehicle. The map of this experiment is shown in Figure 5. The longitude is from 121.5083335 to 121.5677824, and the latitude is from 25.0283751 to 25.0644845. The length of this map is 6000 m, and the width is 4000 m.

### 3.3. SD-QGRID Offline Q-Learning Training Process

The flow of the SD-QGrid offline Q-learning training process is shown in Figure 6. The information of the vehicle environment is handed over to each module of the SDN Control Node (CN) and RSUs, and finally, generates Q-tables of each grid. The steps of the SD-QGRID offline Q-learning training process are described as follows:After the initialization of the SD-QGrid, all vehicles on a road segment issue HELLO messages to the corresponding RSUs.The RSU collects vehicle trajectory information from connected road segments and analyzes it to generate the historical trajectory tables.The RSU periodically transmits the analyzed tables and vehicle trajectories of the connected road segments to the SDN CN.The SDN CN aggregates the historical vehicle information of each grid from the information sent by its RSUs in the network environment.The SDN CN extracts important Q-learning parameters from the aggregated information of each grid. It finally uses Q-learning to train the Q-model for calculating the Q-table and Q-values of each grid.

The following provides a functional description of the modules in the SD-QGrid offline Q-learning training process.

#### 3.3.1. RSU Analysis Unit

HELLO messages periodically issued from all vehicles on a road segment to the corresponding RSUs include their GPS data, speed, moving direction, vehicle IDs, neighbor vehicle IDs, etc. With the following operations, the average number of vehicles in each road segment is counted and the historical trajectory of each vehicle in this road segment is determined by the RSU. This information is cached on the trajectory cache of the RSU.

Each day is divided into eight time periods, as shown in Table 2 [22]. Historical trajectories of vehicles are listed in Table 3, which records the vehicle ID, time stamp and the longitude and latitude values of each trajectory point. The RSU counts the average number of vehicle nodes in its connected road segments in a certain time period within the experimental date range. The calculation method is shown in Equation (3). Based on the vehicle ID and its GPS data, i.e., the current position of the vehicle, in the HELLO message, Table 4 counts the number of vehicles that appear in the connected road segment during this time period every day in the date range, before dividing it by the total number of days in the date range and recording it in the table. An example of a table is Table 4, e.g., 2021-10-13–10-12; the average number of nodes appearing in road segment i of Grid (0,2) over a period of 10:00–12:00 is 314.
(3)$$\bar{N_{i}^{t}}=\frac{\sum_{d\epsilon D}N_{i,\;d}^{t}}{\left|D\right|}$$
where D is the date range, |D| is the total number of days in *D*, $N_{i,\;d}^{t}$ is the number of nodes of road segment i in period t on day d, and $\bar{N_{i}^{t}}$ is the average number of nodes of road segment i in period t. Please note that each road segment is identified by the (x, y) coordinates of both its intersections, which are called the *start* and *end* ones here. Further, multiple HELLO messages issued by the same vehicle only increase this RSU’s counter, i.e., $N_{i,\;d}^{t}$, by one, if this vehicle stays in the same road segment in period t.

If the road segment crosses from a grid into one of the up, down, left, right, upper right, lower left, upper left, or lower right neighbor grids, its RSU located in this grid records the number of outgoing nodes, which moves from this grid to the neighbor one. The average number of outgoing nodes in a certain time period within the experimental date range is calculated by an equation such as Equation (3) and stored in a table, as shown in Table 5. The outgoing direction of this road segment is from the current grid toward the neighbor one.

#### 3.3.2. Aggregation Unit of SDN CN

After each RSU uses its analysis unit to generate Table 4 and Table 5 (if available) for recording the average number of nodes and the average number of outgoing nodes for each connected road segment in period t, respectively, it will periodically forward these tables and the vehicle trajectory in Table 3 to the SDN CN, which stores the vehicle trajectory in its trajectory database. As the SDN CN collects these tables sent by all RSUs in period t, it will adopt its aggregation unit to aggregate the average number of nodes in Table 6 and the average number of outgoing nodes in Table 7 from tables forwarded by all RSUs belonging to each grid. Hence, the SDN CN is able to further calculate the directionality of each grid using the aggregated vehicle information, and count it in a certain period of time within the experimental date range as follows: It determines the total number of outgoing nodes, i.e., $n_{i},\;i=\left\{0,\;1,\;2,\;\ldots ,\;7\right\},$ that each grid moves to the neighbor grid in eight directions in the date range, which is then divided by the total number of days in the date range to calculate the average directionality of individual grids in this period for each day in the date range. It finally records the statistical data in the average grid directionality table, as shown in Table 7.

#### 3.3.3. Learning Unit of the SDN CN

The learning unit of the SDN CN uses Q-learning to calculate the Q-table of each grid based on Table 6 and Table 7, which are aggregated from the information in Table 4 and Table 5 that is sent by RSUs in each grid. The SD-QGrid proposes to record the Q-values of the eight neighbor grids for each grid in the Q-table database. Whenever the learning unit of the SDN CN completes its work, the SD-QGrid offline Q-learning training process ends.

We define Q-learning parameters of SD-QGrid as follows:

(a) State represents the node holding the packet in Grid i at time period t.

(b) Action represents the neighbor grid that is chosen to transmit the packet.

(c) Reward indicates whether the selected neighbor grid is the grid where the destination node is located. If this is true, we will give this state a reward of 100; otherwise, it receives a reward of 0.

The learning unit of the SDN CN performs Q-learning to generate a Q-table of each grid. The flow chart is shown in Figure 7, and the flow description is described as follows:Step 1:After the SDN CN aggregates Table 4 and Table 5 of the corresponding time periods from each RSU of every grid, it calculates the ratio of the average number of nodes in the current grid to the average number of nodes in the map in this time period, according to Table 6. Then, it proceeds to Step 2.Step 2:According to Table 7, the SDN CN calculates the ratio of the average number of outgoing nodes from the current node to each neighbor grid to the sum of the average number of outgoing nodes from the current node to all eight neighbor grids. Then, it proceeds to Step 3.Step 3:From Equation (1), Q-learning can use the discount rate to judge the packet transfer between grids. When the discount rate is higher, it means that the future reward from the next state will be higher. Relatively speaking, when the discount rate is higher, the future reward will be higher. Therefore, the discount rate is set that will be obtained by sending packets to neighboring grids in different directions according to the ratio of the average number of nodes per grid, and the ratio of the average directionality of each grid calculated above. We define the discount rate $\gamma_{t\;}^{k}$ for selecting neighbor $Grid_{k}$ at time period t as follows:(4)$$\gamma_{t\;}^{k}=\left(w\times \frac{H_{t}^{s}}{\sum H_{t}}+j\times \frac{N_{t}^{k}}{\sum_{k}N_{t}^{k}}\right)\times m\;$$
where k=0, 1, 2, …, 7, representing eight directions. $H_{t}^{s}$ is the average number of nodes in $Grid_{s}$ at time period t. $\frac{H_{t}^{s}}{\sum H_{t}}$ is the ratio of the average number of nodes in the current grid to the average number of nodes in the map at time period t at Step 2. $N_{t}^{k}$ is the average number of outgoing nodes in the neighbor grid from $Grid_{s}$ in the direction k at the time period t. $\overset{7}{\underset{k=0}{\sum }}N_{t}^{k}$ is the sum of the average number of outgoing nodes to the neighbor grid in eight directions from $Grid_{s}$ at the time period t. $\frac{N_{t}^{k}}{\sum_{k}N_{t}^{k}}$ is the ratio of the number of outgoing nodes in each direction to the total number of outgoing nodes in all eight directions at Step 3. Parameter *m* means that the value of $\gamma_{t\;}^{k}$ is between 0.3 and 0.9, as to avoid the value of Q-value being too large, which, in turn, affects the choice of the packet transmission route. Finally, w and j are the weighting factors of these two terms in Equation (4), respectively.Step 4:The SDN CN performs the Q-learning calculation with the calculated discount rate to generate the corresponding Q-table of each grid for different time periods, and records the Q-value of each grid for the eight neighbor grids in the Q-table database.

### 3.4. SD-QGrid Online Routing Decision Process

The SD-QGrid has to execute the online routing decision process when a vehicle node $v_{i}$ carrying a packet intends to transmit a certain packet in grid $G_{i}$ to the destination vehicle node $v_{d}$ located in the destination grid $G_{d}$. As the offline process has been completed, each grid has a precalculated Q-table stored in the Q-table database of the SDN CN. Assume each vehicle $v_{i}$ knows its GPS location $\left(X_{i},\;Y_{i}\right)$, its neighbor nodes within a range $\left(V_{0},\;V_{1},\;\ldots ,\;V_{j}\right)$, and vehicle future movement trajectories. As shown in Figure 8, the associated RSU of the packet-carrying node will perform the online routing decision process, which contains the following steps:The packet-carrying node issues a *Neighbor Query* packet to its RSU. This Neighbor Query packet contains vehicle IDs of the packet-carrying node, its neighbor nodes, and the destination node.As the RSU receives the Neighbor Query packet, it stores those vehicle IDs first. Then, it uses them to check whether historical trajectories of neighbor nodes of the packet-carrying node can be found on its trajectory cache and whether Q-tables of the grid where this RSU belongs to and the eight neighbor grids have been cached in it. If yes, it proceeds to Step 3. If not, it will notify the SDN CN to sends back only the missing historical trajectories of neighbor nodes and Q-tables of its associated grid and eight neighbor grids from the trajectory database and Q-table database, respectively. Then, these missing data are cached on the trajectory cache and Q-table cache of the RSU.From the Q-table of this grid, the routing decision unit of the RSU first selects the grid with the largest Q-value among the eight neighbor grids as the best next-hop grid.According to the best next-hop grid, Q-tables of the eight neighbor grids, and historical trajectories of the neighbor nodes of the packet-carrying node, the RSU then executes the two-hop trajectory-based routing (THTR) algorithm to select as the relay the neighbor node whose future driving trajectory continues along the best next-hop grid and the best two-hop next grid.Finally, the RSU issues the *Neighbor Response* packet containing the vehicle ID of the selected neighbor node to inform the packet-carrying node to forward the packet to this neighbor node.

Hence, THTR finds a better packet transmission route, improves the packet delivery ratio, and reduces the end-to-end delay in the V2X environment.

Figure 9 shows that the best next-hop grid of $Grid_{A}$ is $Grid_{A,1}$ and the two-hop next-grids of $Grid_{A,1}$ are $Grid_{A,1.1}$ and $Grid_{A,1.2}$. If a node’s driving trajectory passes through the best next-hop grid and the best two-hop next grid in the left picture, i.e., from $Grid_{A,1}$ to $Grid_{A,1.1}$, which is more in line with the overall directionality of packet transmission from the start point $v_{s}$ to the end point $v_{d}$, then this node is a better next-hop packet-forwarding node. Conversely, if the driving trajectory of a node passes through the best next-hop grid $Grid_{A,1}$ and the two-hop next grid $Grid_{A,1.2}$, which is not the best two-hop next grid of $Grid_{A,1}$ in the right figure, it can be seen that the vehicle node deviates from the overall directionality of the packet transmission, as compared to the vehicle on the left. Hence, this vehicle is not a good relay node.

#### 3.4.1. The Link Expiration Time (LET) between Two Nodes

Additionally, we use the location prediction mechanism to calculate the *link expiration time* (LET) [31] of the adjacent vehicle, that is, the possible connection time between the current node and the adjacent node. If the calculated LET is not smaller than the minimum time to completely forward a packet from the current node to the adjacent node, this adjacent node is considered as an eligible candidate relay node in the online routing decision process. It is assumed that the clocks of all vehicle nodes in the network are synchronized. At a certain point in time, let (*x_i_*, *y_i_*) and (*x_j_*, *y_j_*) be the two-dimensional coordinate positions of vehicles *i* and *j*. *θ_i_*, *θ_j_* (0 ≤ *θ_i_*, *θ_j_* < 2π) and *v_i,_ v_j_* are the moving directions and moving speeds of *i* and *j*, respectively, and r is the transmission range of wireless signals between vehicle nodes. We can use Equation (5) to calculate the *LET_ij_* that vehicles *i* and *j* can maintain their connections because both of them stay within the wireless communication range:(5)$$LET_{ij}=\frac{-(ab+cd)+\sqrt{(a^{2}+c^{2})r^{2}-{\left(ad-bc\right)}^{2}}}{a^{2}+c^{2}}$$
where:

*a* = *v_i_* cos *θ_i_* − *v_j_* cos *θ_j_*

*b* = *x_i_* − *x_j_*

c = *v_i_* sin *θ_i_* − *v_j_* sin *θ_j_*

*d* = *y_i_* − *y_j_*

if *v_i_* = *v_j_* and *θ_i_* = *θ_j_*, *LET_ij_* = ∞.

#### 3.4.2. Routing Decision Unit of RSU

As mentioned above, the routing decision unit of the RSU first selects the grid with the largest Q-value among the eight neighbor grids as the best next-hop grid from the Q-table of this grid. According to the best next-hop grid, Q-tables of the eight neighbor grids, and historical trajectories of the neighbor nodes of the packet-carrying node, the RSU then executes the two-hop trajectory-based routing (THTR) algorithm to select the neighbor node as the relay whose future driving trajectory continues along the best next-hop grid and the best two-hop next grid.

The flow of the SD-QGrid routing decision process, shown in Figure 10, is listed as follows:

Step 1:Determine whether the packet-carrying node $v_{i}$ is the destination node. If so, go to Step 9; if not, go to Step 2.Step 2:Determine whether $v_{i}$ has any neighbor node. If so, go to Step 3; if not, go to Step 8.Step 3:For some neighbor node $v_{j}$ of $v_{i}$ located in the neighbor grid $G_{j}$, test whether the Q-value $Q_{j}$ of the neighbor grid $G_{j}$ is higher than the Q-value $Q_{i}$ of grid $G_{i}$, where the vehicle $v_{i}$ is located. If so, go to Step 4; if not, go to Step 6.Step 4:If the moving direction of the neighbor node is known and it has a fixed route and schedule (such as a bus), go to Step 5; if not, go to Step 6.Step 5:Execute Algorithm 1. If the best next-hop node $v_{k}$ exists, go to Step 7; if not, go to Step 6.Step 6:Among all neighbor nodes in the neighbor grid with the higher Q-value than that of the current grid, select the neighbor node $v_{k}$ that is closest to the destination. If $v_{k}$ exists, go to Step 7; if not, go to Step 8.Step 7:Notify the packet-carrying node to use this node as the next-hop node and forward the packet to it. Go to Step 1.Step 8:Vehicle $v_{i}$ continues to hold the packet and waits for the next transmission opportunity. Go to Step 1.Step 9:If the current node is the destination node, the entire routing decision process ends.

**Algorithm 1.** Next-hop node selecting algorithm.Input:
The node *A* currently carrying the packet, the destination node *D* of the packet, and the coordinate position of each node at each time point.Q-table, which is used for the overall network environment.$\mathrm{The}\;\mathrm{set}\;N_{A}$ formed by all neighbor nodes *B* of *A*$,\;\forall B\in N_{A}$.
Output:
Select node *B*, which is the best next-hop neighbor of node *A*, and send the packet to *B*; or, node *A* continues to carry the packet.
Definition:
The Q-value from $Grid_{A}$ to $Grid_{B}$ is $Qvalue_{B}^{A}\;$($Grid_{A}$ is adjacent to $Grid_{B}$).The grid where node *x* locates is $Grid_{X}$.
1$TwoHopVal_{A}=TwoHopVal_{B}=OneHopVal_{A}=OneHopVal_{B}=0$2Let $Grid_{A}$ be the grid of packet-carrying node *A* and $Grid_{D}$ be the grid of the destination node *D*.3Query the SDN CN to retrieve the Q-table of $Grid_{A}$ and those of neighbor grids. Store them in the Q-table cache of the RSU.4Calculate the next-hop grids with the top three Q-values in the Q-table of $Grid_{A}$:$\;Grid_{1,1}$, $Grid_{1,2}$, $Grid_{1,3}$.5for t = (1, 2, 3):
$\;\mathrm{if}\;Grid_{1,t}$ exists, then calculate its next-hop grid with the top three Q-values of $Grid_{1,t}$$:Grid_{1,t,1}$$,\;Grid_{1,t,2}$$,\;Grid_{1,t,3}$. // the two-hop next grids.
6For all neighbor nodes *B* of *A*:7If node *B* is the destination node *D*, send the packet to *B*, which ends the online packet routing process.8Otherwise, calculate $LET_{B}^{A}$. If $LET_{B}^{A}$ is less than the time $T_{p}$ required to deliver a packet, node B is not considered as an eligible candidate relay node.9Call Algorithm 2, i.e., One-Two-Hop Value *(A, B*)*,* to obtain $TwoHopVal_{B}$ and $OneHopVal_{B}$.10The end of the loop.11$TwoHopVal_{\;}$= Max{$TwoHopVal_{B},\;\forall B\in N_{A}\}$12$OneHopVal_{\;}$=$\;Max\left\{OneHopVal_{B},\forall B\in N_{A}\right\}$13If ($TwoHopVal_{\;}>0$)
Select the node with the largest TwoHopVal as the best next-hop node.
14else15Select the node with the maximum OneHopVal as the best next-hop node

Algorithm 1 uses the Q-value to calculate up to nine (next-hop grid, two-hop next grid) combinations based on the grid $Grid_{A}$ where the current packet is located. It then selects the two-hop next grid combination, depending on the next-hop grids with the top three Q-values in line 5. For all neighbor nodes *B* of *A*, Algorithm 1 uses the link expiration time of adjacent vehicles to filter the candidate relay nodes in line 8. It calls Algorithm 2, i.e., the One-Two-Hop Value *(A, B*)*,* in line 9 to obtain $TwoHopVal_{B}$ and $OneHopVal_{B}.$ Lines 11 and 12 find out which neighbor node owns the maximum TwoHopVal and OneHopVal values among all neighbor nodes, respectively. The vehicle with the highest value is selected as the relay node for packet forwarding. In addition, we preferentially select vehicle nodes whose trajectories pass through both the next-hop grid and the two-hop next grid in line 13, and then, select vehicle nodes that only pass through the next-hop grid in line 15 for packet forwarding.

**Algorithm 2.** One-Two-Hop Value (Node A, Node X)Input:
The three next-hop grids, i.e., $Grid_{1,1}$, $Grid_{1,2}$, and $Grid_{1,3}$ of $Grid_{A}$.The three two-hop next grids, i.e., $Grid_{1,1,1}$, $Grid_{1,1,2},$ and $Grid_{1,1,3}$ of $Grid_{1,1}$; $Grid_{1,2,1}$, $Grid_{1,2,2},$ and $Grid_{1,2,3}$ of $Grid_{1,2}$; $Grid_{1,3,1}$, $Grid_{1,3,2}$, and $Grid_{1,3,3}$ of $Grid_{1,3}$.The next-hop candidate node x of node *A*.
Output:
$TwoHopVal_{X}$$OneHopVal_{X}$
Definition:
$TimeDiff_{X\to X^{’}}^{X}$ is the difference between the time when node X first enters $Grid_{X}$ and the time when x first enters $Grid_{X^{’}}$, according to its trajectory. Please note this notation indicates that node x does not pass through any other grid between $Grid_{X}$ and $Grid_{X^{’}}$. If $Grid_{X}=Grid_{X^{’}}$, then the value of $TimeDiff_{X\to X^{’}}^{X}$ is the minimum value 1. If node X does not pass through $Grid_{X^{’}}$ in its trajectory, then the value of $TimeDiff_{X\to X^{’}}^{X}$ is 0.
1for t = (1, 2, 3) //Time difference between the current time in the current grid and the first position in the next-hop grid.$\mathrm{If}\;Grid_{X}=Grid_{A}$$,\;\mathrm{calculate}\;TimeDiff_{X\to 1,t}^{X}$$\;\mathrm{of}\;\mathrm{node}\;X\;\mathrm{from}\;Grid_{X}\;$$\mathrm{to}\;Grid_{1,t}$$\mathrm{If}\;Grid_{X}=Grid_{1,t}$$,\;TimeDiff_{X\to 1,t}^{X}=1$$.\;//\mathrm{node}\;x\;\mathrm{in}\;\mathrm{the}\;\mathrm{next}-\mathrm{hop}\;\mathrm{grid}\;Grid_{1,t}$.for s = (1, 2, 3) //Calculate the time difference between the current time in the current grid and the first position in the two-hop next grid.$\mathrm{If}\;Grid_{X}=Grid_{1,t,s}$$,\;TimeDiff_{X\to 1,t,s}^{x}=1$$.\;//\mathrm{node}\;x\;\mathrm{in}\;\mathrm{the}\;\mathrm{two}-\mathrm{hop}\;\mathrm{next}\;\mathrm{grid}\;Grid_{1,t,s}$.else //node x in the next-hop grid.
$\mathrm{Calculate}\;TimeDiff_{X\to 1,t\to 1,t,s}^{x}$$\;\mathrm{of}\;\mathrm{node}\;x\;\mathrm{from}\;Grid_{X}$ to $Grid_{1,t,s}$ via $Grid_{1,t}$$.\;//\;\mathrm{node}\;x\;\mathrm{first}\;\mathrm{moves}\;\mathrm{to}\;Grid_{1,t}$, and then to $Grid_{1,t,s}.$$TimeDiff_{1,t\to 1,t,s}^{x}$$=TimeDiff_{X\to 1,t\to 1,t,s}^{x}-TimeDiff_{X\to 1,t}^{x}$//$TimeDiff_{1,t\to 1,t,s}^{x}$$\;\mathrm{of}\;\mathrm{node}\;x\;\mathrm{is}\;\mathrm{the}\;\mathrm{TimeDiff}\;\mathrm{from}\;Grid_{1,t}$$\;\mathrm{to}\;Grid_{1,t,s}$.
2dst_grid = false3for t = (1, 2, 3) //calculate the one-hop weight $V_{X\to 1,t}^{X}$ when node x is in the next-hop grid $Grid_{1,t}$.
if $TimeDiff_{X\to 1,t}^{X}>0$: $V_{X\to 1,t}^{X}$ = [$\;\frac{1}{Distance\left(X,\;D\right)}\times \frac{\;Qvalue_{1,t}^{A}}{TimeDiff_{X\to 1,t}^{X}}\;$]if $Grid_{1,t}$ is the grid where the destination node is located:
$V_{X\to 1,t}^{X}=V_{X\to 1,t}^{X}\;*100$ //let the weight $V_{X\to 1,t}^{X}$ become much higher.dst_grid = truefor s = (1, 2, 3) //calculate the one-hop weight $V_{X\to 1,t,s}^{X}$$\;\mathrm{when}\;\mathrm{node}\;x\;\mathrm{is}\;\mathrm{in}\;\mathrm{the}\;\mathrm{two}-\mathrm{hop}\;\mathrm{next}\;\mathrm{grid}\;Grid_{1,t,s}.$
if $TimeDiff_{X\to 1,t,s}^{X}>0$: $V_{X\to 1,t,s}^{X}$ = [$\;\frac{1}{Distance\left(X,\;D\right)}\times \frac{\;Qvalue_{1,t,s}^{A}}{TimeDiff_{X\to 1,t,s}^{X}}\;$]if $Grid_{1,t,s}$ is the grid where the destination node is located:
$V_{X\to 1,t,s}^{X}=V_{X\to 1,t,s}^{X}\;*100$dst_grid = true4$OneHopVal_{X\;}$=$\;Max\;\left\{V_{X\to 1,t}^{X},\;V_{X\to 1,t,s}^{X}\right\}$ // the maximum weight among the one-hop weight $V_{X\to 1,t}^{X}$ and the one-hop weight $V_{X\to 1,t,s}^{X}$.5If dst_grid = false // calculate the two-hop weight.
for t = (1, 2, 3)
for s = (1, 2, 3)
$\mathrm{if}\;TimeDiff_{X\to 1,t}^{X}\;\mathrm{and}\;TimeDiff_{1,t\to 1,t,s}^{X}>0$:
$V_{X\to 1,t\to 1,t,s}^{X}\;$$=[\frac{1}{Distance\left(X,\;D\right)}\times \frac{\;Qvalue_{1,t}^{A}}{TimeDiff_{X\to 1,t}^{X}}\times \frac{\;Qvalue_{1,t,s}^{1,t}}{TimeDiff_{1,t\to 1,t,s}^{X}}\;$]If $Grid_{1,t,s}$ is the grid where the destination node is located:
$V_{X\to 1,t\to 1,t,s}^{X}=V_{X\to 1,t\to 1,t,s}^{X}\;*100$


6$TwoHopVal_{X\;}$= Max{$V_{X\to 1,t\to 1,t,s}^{X}$} // the maximum weight among all two-hop next grid weight $V_{X\to 1,t\to 1,t,s}^{X}$.7Return $OneHopVal_{X}$, $TwoHopVal_{X\;};$


For example, as shown in Figure 11, the source vehicle $v_{s}$ located in $Grid_{A}$ carries the packet in the beginning. First, the associated RSU of $v_{s}$ queries the SDN CN to retrieve the Q-table of $Grid_{A}$ and those of neighbor grids, which are stored in the Q-table cache of the RSU. Then, the routing decision unit of the RSU finds the next-hop grids, i.e., $Grid_{1,1}$, $Grid_{1,2}$, and $Grid_{1,3}$, with the top three large Q-values in the Q-table of $Grid_{A}$. As shown in Figure 12, take the next-hop grid $Grid_{1,1}$ as an example. The routing decision unit of the RSU further queries the Q-table of $Grid_{1,1}$ to find the next-hop grid of $Grid_{1,1}$ with the top three large Q-values of $Grid_{1,1,1}$, $Grid_{1,1,2}$, and $Grid_{1,1,3}$, which constitute the two-hop next grid of $Grid_{A}$.

Algorithm 2 refers to the next-hop grid and the two-hop next grid combinations calculated by Algorithm 1, and calculates TimeDiff values of these nine combinations for each neighbor node X of A at time *t*. There are three kinds of grids where X may stay: either at the same grid $Grid_{A}$, the next-hop grid $Grid_{1,t}$ or the two-hop next grid $Grid_{1,t,s}$ of A. The calculation method is shown in Figure 13. Algorithm 2 calculates $TimeDiff_{X\to 1,1}^{X}$, which is the time difference between the current time *t* when node X stays in $Grid_{X}$ and the time when X first enters the next-hop grid $Grid_{1,1}$, depending on X’s trajectory. Here, $Grid_{1,1,1}$ is one of the two-hop next grids of $Grid_{1,1}$. Hence, $TimeDiff_{1,1\to 1,1,1}^{X}$ is the time difference between the time when X first enters the next-hop grid $Grid_{1,1}$ and the time when X first enters the two-hop next grid $Grid_{1,1,1}$. After TimeDiff values from $Grid_{X}$ to all next-hop grids and two-hop next grids have been calculated in line 1, line 3 of Algorithm 2 calculates the one-hop weight $V_{X\to 1,t}^{X}$ when node X is in the next-hop grid $Grid_{1,t}$, or the one-hop weight $V_{X\to 1,t,s}^{X}$ when node X is in the two-hop next grid $Grid_{1,t,s}$. The weight value $V_{X\to G}^{X}$ is proportional to the Q-value, i.e.,$\;Qvalue_{G}^{A}$ queried from the Q-table of $Grid_{A}$ and of $Grid_{G}$ where X stays, but inversely proportional to the Euclidean distance Distance(X, D) between node X and destination node D, and the TimeDiff value from $Grid_{X}$ to $Grid_{G}$. Line 4 finds $OneHopVal_{X}$, which is the maximum weight among all one-hop weights. Similarly, line 5 calculates the two-hop weight $V_{X\to 1,t\to 1,t,s}^{X}$ when node X is in the next-hop grid $Grid_{1,t}$ and will directly move to the two-hop next grid $Grid_{1,t,s}$. The weight value $V_{X\to 1,t\to 1,t,s}^{X}$ is proportional to the Q-value, i.e.,$\;Qvalue_{1,t}^{A}$ queried from Q-table of $Grid_{A}$, and $Qvalue_{1,t,s}^{1,t}$ queried from the Q-table of $Grid_{1,t}$, but is inversely proportional to the Euclidean distance Distance(X, D) between node X and destination node D, $TimeDiff_{X\to 1,t}^{X}$ from $Grid_{X}$ to $Grid_{1,t}$, and $TimeDiff_{1,t\to 1,t,s}^{X}$ from $Grid_{1,t}$ to $Grid_{1,t,s}$. Line 6 finds $TwoHopVal_{X}$, which is the maximum weight among all two-hop weights. Finally, line 7 ends Algorithm 2 by returning $OneHopVal_{X}$ and $TwoHopVal_{X}$ to Algorithm 1.

## 4. Performance Evaluation

### 4.1. Simulation Environment and Parameters

In this section, we use real vehicle trajectories to evaluate the performance of our proposed SD-QGrid method, and compare it with three other methods. We use Python for reinforcement learning calculations and NS-3 [32] for a network simulation experiment. The MAC protocol in NS-3 is IEEE 802.11p and the radio propagation model is the Log Distance Propagation Loss Model in this simulation.

We capture the 4000-by-6000-m Taipei City area as the simulated environment, as shown in Figure 5, and use the bus and vehicle nodes within the range as the packet-forwarding nodes. Then, we add fixed node pairs as source and destination nodes on random roads in the map. The distance between the fixed node pairs is not less than 4000 m, and fixed nodes cannot be used as transit nodes for packet routing. The experimental parameters are shown in Table 8, where the number of fixed nodes is numPair, the vehicle node broadcasts a HELLO packet after a beacon time, and the fixed node generates a message packet after a message time. The parameters of the simulation experiment are the transmission range, TTL, numPair, and message time.

We compare the following three routing algorithms with the SD-QGrid method:QGrid_G [16]: A grid-based routing protocol proposed by QGrid [16]; its intergrid routing uses the greedy method to find the nearest next-hop neighbor node to the destination.Advanced QGrid [16]: QGrid improves the routing protocol for vehicle nodes with relatively fixed trajectories, such as buses. If the vehicle node passes through the grid where the packet destination is located in the future, it will continue to carry the packet until it enters the grid, and then proceed with next-hop routing selection to transfer the packet.GPSR [33]: A position-based routing protocol that continuously forwards packets from the nearest neighbor to the destination until it reaches the destination.

We use the different transmission ranges, TTLs, numPairs, and message times, as shown in Table 8, as the horizontal axis, and use the following indicators to compare the performance of various routing algorithms. These figures are drawn with a 95% confidence interval. The number of seeds used in NS-3 is five.

(a)Delivery ratio: the ratio of the successful arrival of message packets to the total number of message packets generated.(b)Average end-to-end delay: how long it takes, on average, for a message packet to travel from the source to the destination.(c)Overhead: the ratio of the total number of forwarded message packets to the total number of originally sent message packets.

### 4.2. Simulation Results

First, we compare the impact of the transmission range (distance) on the performance of the routing algorithm. Figure 14 shows that the distance gradually increases, and the transmission success rate also increases. On the contrary, because the packet arrives at the end point faster, the end-to-end delay and the overall network environment overhead will be reduced accordingly. Among the four methods, the GPSR method simply selects the neighbor node closest to the end point for packet forwarding; thus, the end-to-end delay is the lowest, but the transmission success rate is also the lowest, and the overhead is also the highest. The advQGrid transmission success rate and overhead are better than that of QGrid because the vehicle node will continue to carry packets until arrival at the destination grid, but in exchange for the highest end-to-end delay. On the contrary, although our proposed SD-QGrid method considers the real vehicle movement trajectory and the source-to-destination directionality in different time periods, compared with QGrid and its improved method advQGrid, it pays slightly more overhead, but achieves the highest delivery ratio and considerably reduces its end-to-end delay. In conclusion, compared with QGrid, our method increases the average delivery ratio by 17%, reduces the end-to-end delay by 27.09%, and pays a 1.04% overhead, which is shown in Table 9. Compared with advQGrid, the average delivery ratio increases by 10.22%, the end-to-end delay decreases by 31.93%, and the overhead is 1.24%.

We compare the impact of the packet time-to-live (TTL) on the performance of the routing algorithm. Figure 15 shows that the packet time-to-live (TTL) gradually increases, the transmission success rate increases accordingly, and the end-to-end delay and the overall network environment overhead also increase. Compared with QGrid and its improved method, advQGrid, although our proposed SD-QGrid method pays slightly more overhead, it can achieve the highest transmission success rate and minimize the end-to-end delay. As the TTL gradually increases, compared with QGrid and advQGrid, the end-to-end delay of SD-QGrid increases more smoothly. In conclusion, compared with QGrid, our method increases the average delivery ratio by 21.01%, reduces the end-to-end delay by 24.7%, and pays a 1.09% overhead, which is shown in Table 10. Compared with advQGrid, the average delivery ratio increases by 13.66%, the end-to-end delay decreases by 34%, and the overhead is 1.53%.

Then, we compare the effect of the fixed number of nodes (numPair) on the performance of the routing algorithm. As shown in Figure 16, as the numPair gradually increases, it means that the total number of packets in the overall network increases, and the transmission success rate and overhead of the four methods decrease accordingly. However, the end-to-end delay is relatively stable. Although our proposed SD-QGrid method pays slightly more overhead than QGrid and its improved advQGrid method, it achieves the highest transmission success rate and the lowest end-to-end delay. In conclusion, compared with QGrid, our method increases the average delivery ratio by 13.66%, reduces the end-to-end delay by 27.67%, and pays a 0.51% overhead, which is shown in Table 11. Compared with advQGrid, the average delivery ratio increases by 9.27%, the end-to-end delay decreases by 35.26%, and the overhead is 1.01%.

Finally, we compare the effect of the packet generation interval (message time) on the performance of the routing algorithm. As shown in Figure 17, as the message time gradually increases, the number of packets in the overall network decreases, and the transmission success rates of the four methods increase. The overhead decreases, and the end-to-end delay remains relatively stable. As shown in the data graphs of the three horizontal-axis parameters mentioned above, compared with QGrid and its improved advQGrid method, our proposed SD-QGrid method pays some extra overhead, but still achieves the highest transmission success rate, and also has the greatest degree of end-to-end delay reduction. In conclusion, compared with QGrid, our method increases the average delivery ratio by 16.47%, reduces the end-to-end delay by 28.65%, and pays a 1.42% overhead, which is shown in Table 12. Compared with advQGrid, the average delivery ratio increases by 10.77%, the end-to-end delay decreases by 35.08%, and the overhead is 2.31%.

## 5. Conclusions and Future Directions

In this study, we address the shortcomings of the traditional V2X grid-based routing algorithm using reinforcement learning, propose the SD-QGrid routing platform, deploy the SDN CN for centralized control, and consider the real vehicle movement trajectory in different time periods. Additionally, regarding the directionality from source to destination, we propose the SD-QGrid offline Q-learning training process to use Q-learning to calculate the Q-table of the eight neighbor grids. This paper further proposes the online routing decision process with the two-hop trajectory-based routing (THTR) packet-routing algorithm, which selects the best next-hop node to forward packets in an intragrid mode. Finally, we use the real vehicle trajectory data in Taipei to conduct simulation experiments in order to draw the data graph of four horizontal-axis parameters, which proves that compared with QGrid and advQGrid, our proposed SD-QGrid method can improve the performance with less than 2% overhead. The transmission delivery ratio is more than 10%, and the end-to-end delay is reduced by more than 25%.

This research is only limited to consider types of vehicles with accurate trajectories, relatively fixed arrival times, and travel paths, such as buses. Hence, we plan to design a new routing architecture to handle packet routing for vehicles without historical trajectory information in the future. We will use a CNN (Convolutional Neural Network) and LSTM (Long Short-Term Memory) to establish a trajectory prediction model. This system will include a packet-forwarding decision model based on Deep Q-Learning. Through the predicted vehicle trajectory and the packet-forwarding decision model, this system can select a better next-hop node to forward packets.

## Figures and Tables

**Figure 3 sensors-22-08222-f003:**
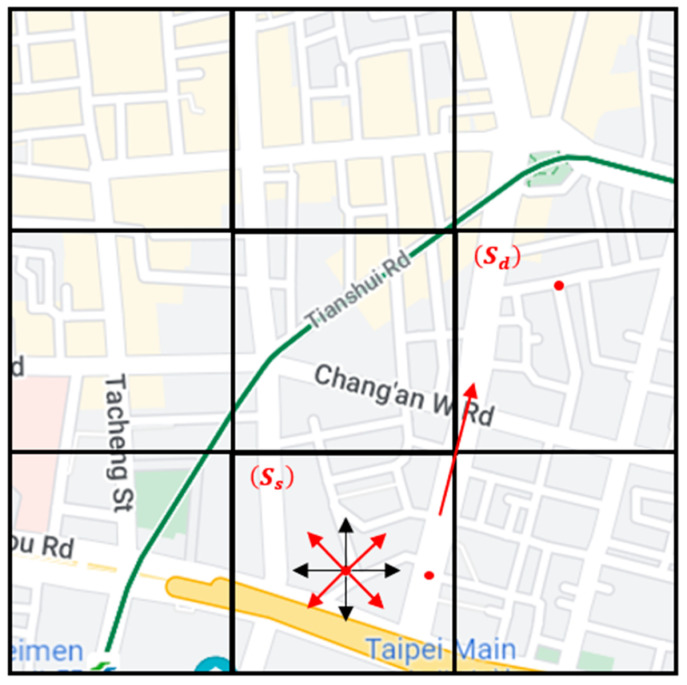
Nodes moving between the current grid and eight neighboring grids in SD-QGrid.

**Figure 4 sensors-22-08222-f004:**
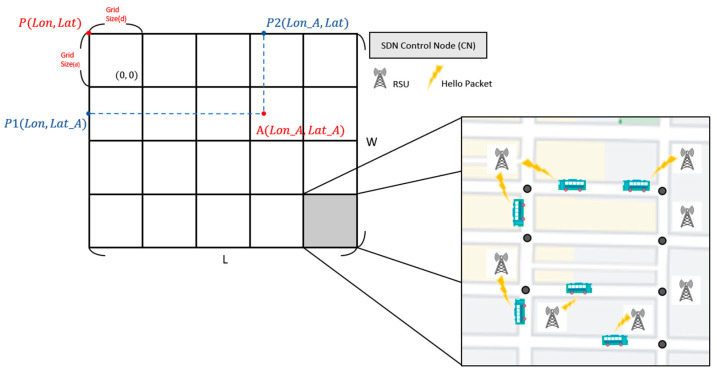
SD-QGrid network architecture.

**Figure 5 sensors-22-08222-f005:**
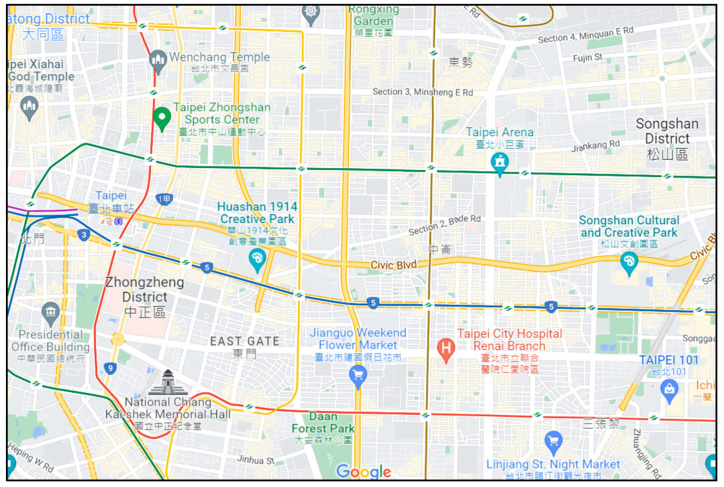
SD-QGrid experimental map.

**Figure 6 sensors-22-08222-f006:**
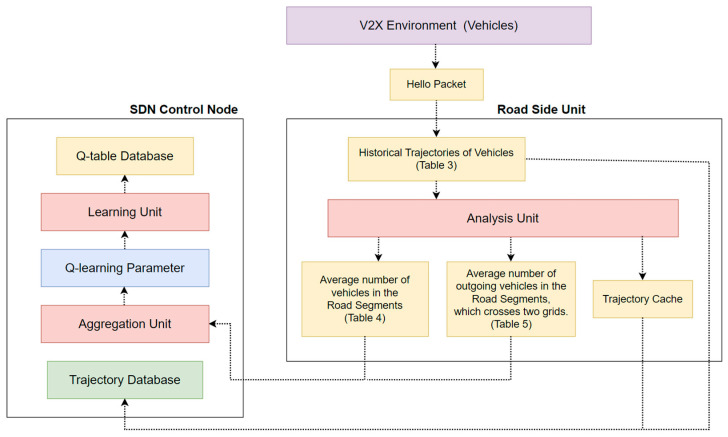
The offline Q-learning training process of SD-QGrid.

**Figure 7 sensors-22-08222-f007:**
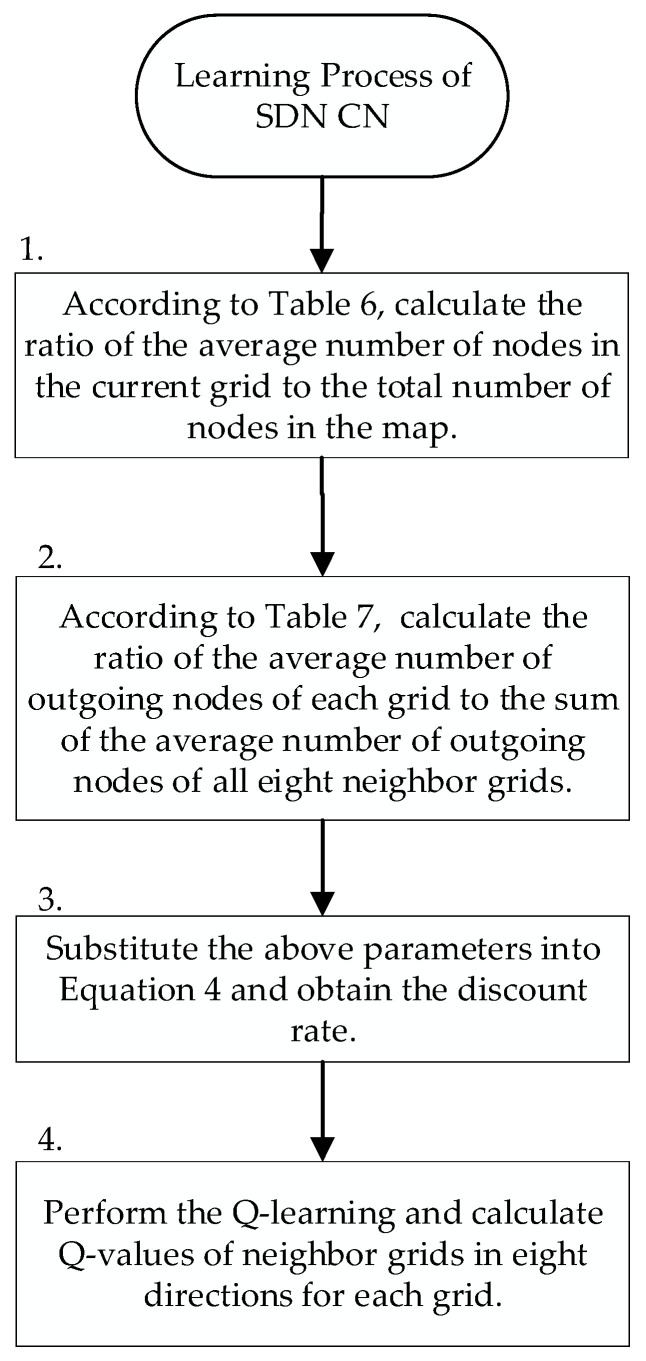
SD-QGrid learning process executed by the SDN CN.

**Figure 8 sensors-22-08222-f008:**
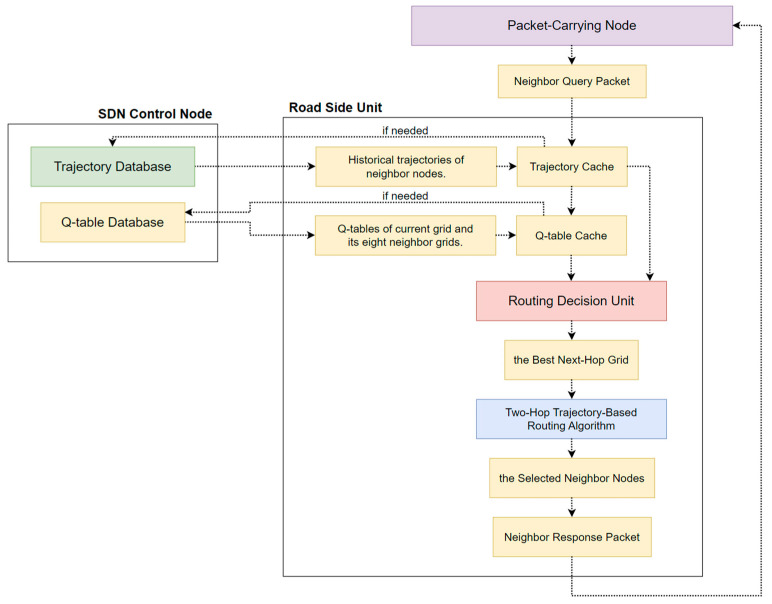
The online routing decision process of SD-QGrid.

**Figure 9 sensors-22-08222-f009:**
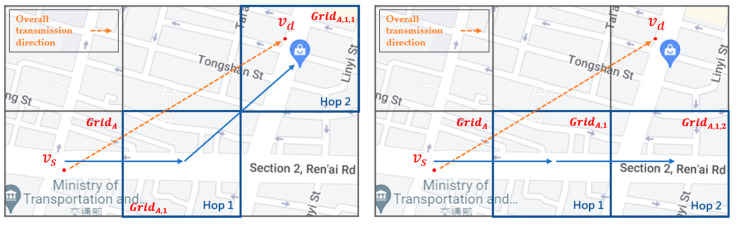
An example of the two-hop next grid in SD-QGrid.

**Figure 10 sensors-22-08222-f010:**
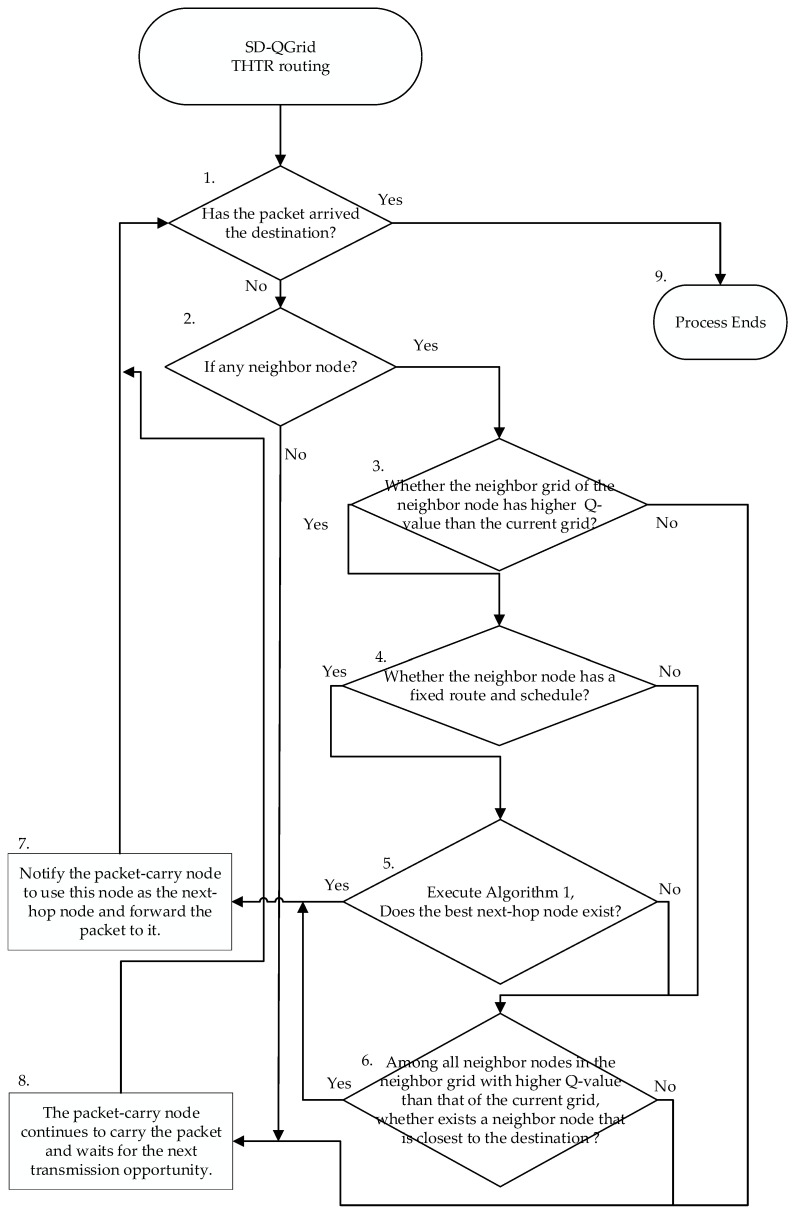
Flow of the SD-QGrid online routing decision process.

**Figure 11 sensors-22-08222-f011:**
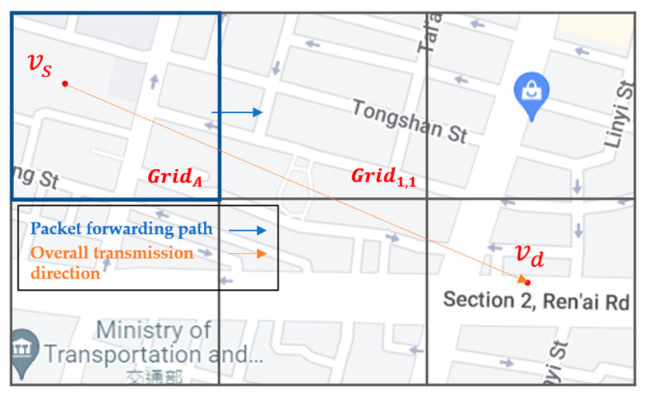
The routing decision unit of RSU selects the next-hop grids of $Grid_{A}$.

**Figure 12 sensors-22-08222-f012:**
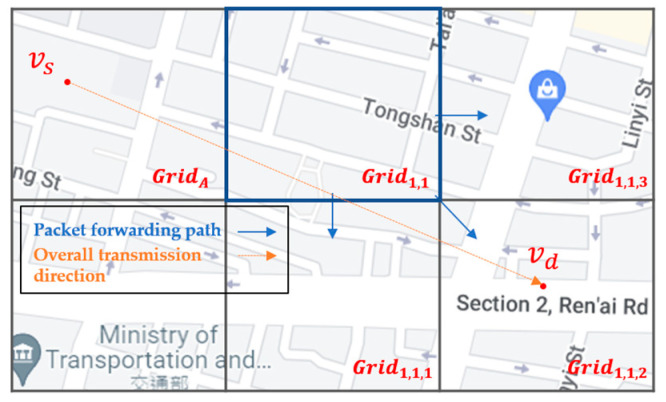
The routing decision unit of RSU further selects the two-hop next grid, using the next-hop grids.

**Figure 13 sensors-22-08222-f013:**
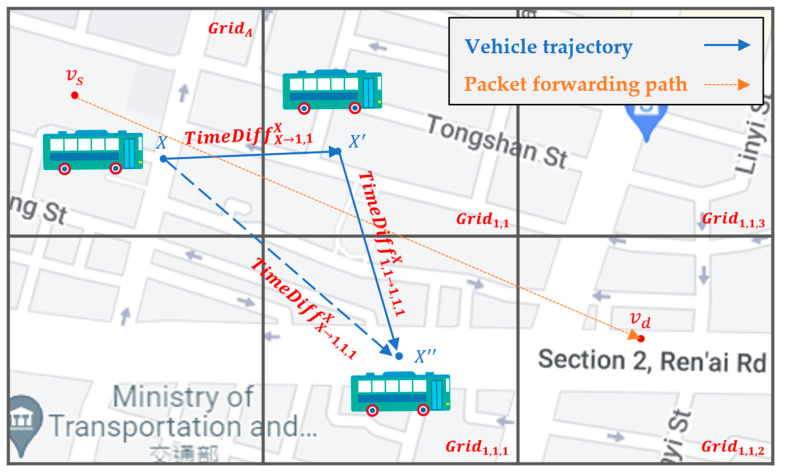
One-Two-Hop Value algorithm.

**Figure 14 sensors-22-08222-f014:**
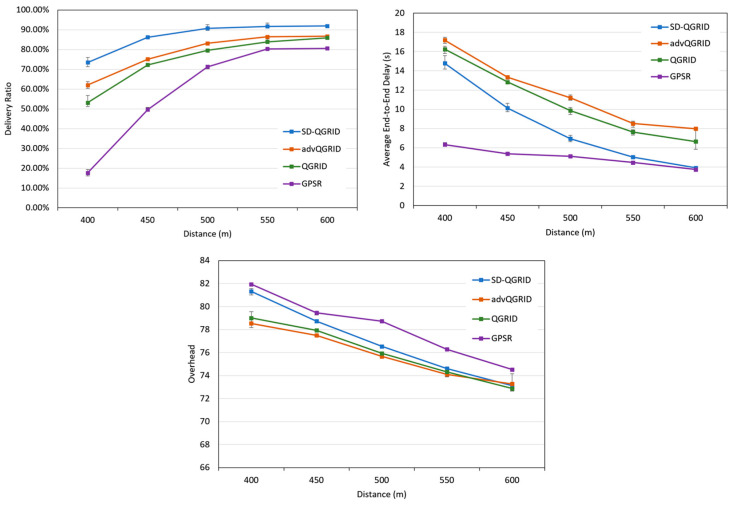
Simulation comparison among different routing protocols vs. different distances.

**Figure 15 sensors-22-08222-f015:**
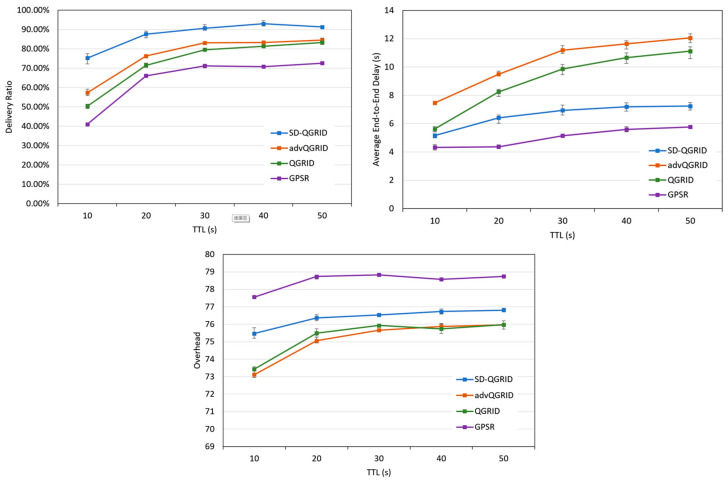
Simulation comparison among different routing protocols vs. different TTLs.

**Figure 16 sensors-22-08222-f016:**
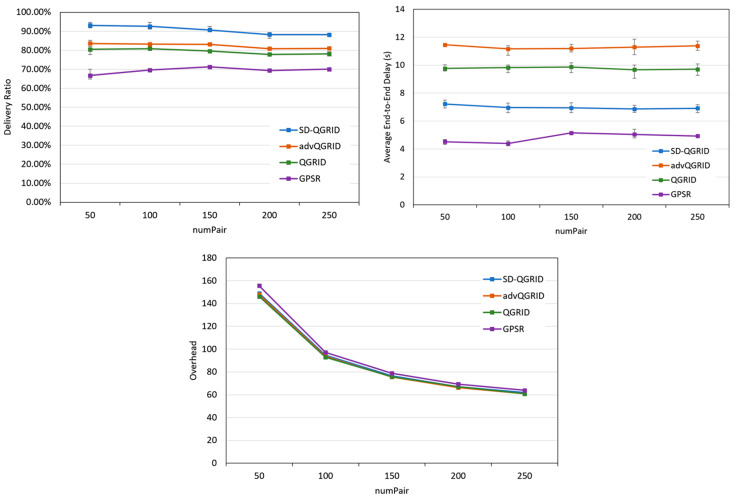
Simulation comparison among different routing protocols vs. different numPairs.

**Figure 17 sensors-22-08222-f017:**
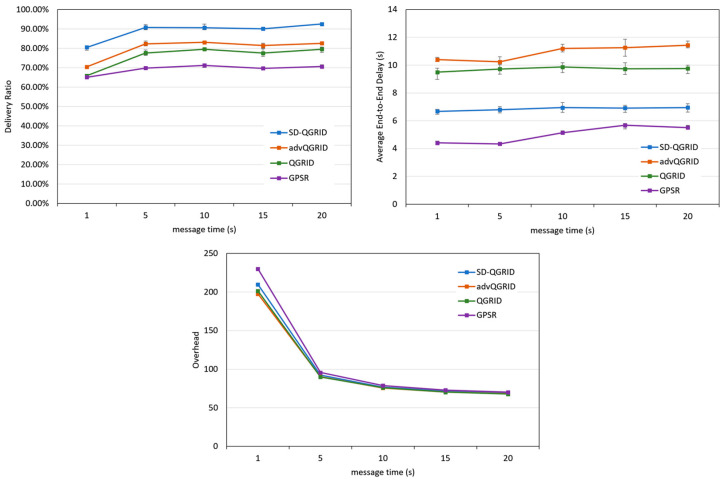
Simulation comparison among different routing protocols vs. different message times.

**Table 1 sensors-22-08222-t001:** Comparison of related studies.

Protocol	Analyzes Real Vehicle Trajectory Data	Considers the Directionality of the Vehicle Trajectory	Considers the Directionality of Source to Destination	Considers the Number of Vehicles in Each Area	Uses Reinforcement Learning for Routing Decisions	Simulation Experiments Using Real Vehicle Trajectories	Considers Two-Hop Next-Grid Routing	Designs Software-Defined Routing Platform
ITAR-FQ	No	No	No	Yes	Yes	No	No	No
ADOPEL	No	Yes	No	Yes	Yes	No	No	No
QTAR	No	No	No	Yes	Yes	No	No	No
QGrid_G	No	No	No	Yes	Yes (only one Q-table, four neighbor grids)	Yes	No	No
AdvQGrid	Yes	No	No	Yes	Yes (only one Q-table, four neighbor grids)	Yes	No	No
SD-QGrid	Yes	Yes	Yes	Yes	Yes (different Q-tables corresponding to different time periods, eight neighbor grids)	Yes	Yes	Yes

**Table 2 sensors-22-08222-t002:** Time period state interval.

Time Period	Congestion Level
23:00–07:00	normal
07:00–09:00	congested
09:00–12:00	normal
12:00–13:00	congested
13:00–17:00	normal
17:00–19:00	congested
19:00–22:00	normal
22:00–23:00	congested

**Table 3 sensors-22-08222-t003:** Historical trajectories of vehicles.

Vehicle ID	Time Stamp	Location Lon	Location Lat
0	weekday_07:00:00 + 08:00,	121.50853	25.04262
0	weekday_07:00:20 + 08:00,	121.5328	25.034475
0	weekday_07:00:40 + 08:00	121.533667	25.044598
…	…	…	…
0	weekday_08:20:20 + 08:00	121.525173	25.052125

**Table 4 sensors-22-08222-t004:** Average number of nodes of road segment i in period t.

**Grid Index**	The Start Intersection Coordinate (x, y) of Road Segment *i*	The End Intersection Coordinate (x, y) of Road Segment *i*	$$\mathrm{Average}\;\mathrm{Number}\;\mathrm{of}\;\mathrm{Nodes}\;\bar{N_{i}^{t}}$$	Time Period t
(0, 2)	(*x_s_*, *y_s_*)	(*x_e_*, *y_e_*)	314	2021-10-13–10-12

**Table 5 sensors-22-08222-t005:** Average number of outgoing nodes of road segment i in period t, if road segment i crosses two grids.

**Current Grid Index**	Neighbor Grid Index	The Start Intersection Coordinate (x, y) of Road Segment *i*	The End Intersection Coordinate (x, y) of Road Segment *i*	Average Number of Outgoing Nodes	Time Period t
(0, 2)	(0, 3)	(*x_s_*, *y_s_*)	(*x_e_*, *y_e_*)	110	2021-10-13–10-12

**Table 6 sensors-22-08222-t006:** Average number of nodes of $Grid_{s}$ in period t.

Grid Index	$$\mathrm{Average}\;\mathrm{Number}\;\mathrm{of}\;\mathrm{Nodes}\;H_{t}^{s}$$	Time Period t
(0, 2)	314	2021-10-13–10-12

**Table 7 sensors-22-08222-t007:** Average grid directionalities of $Grid_{s}$ in period t.

Grid Index	Average Grid Directionality for Eight Neighbor Grids $$N_{t}^{0}/N_{t}^{1}/\ldots /N_{t}^{7}$$	Time Period t
(0, 2)	134/143/506/197/10/5/53/5	2021-10-13–10-12

**Table 8 sensors-22-08222-t008:** Simulation parameters.

Parameters	Parameter Value or Range
α	0.8
γ	[0.3, 0.9]
m	0.1
w	3
j	1
Reward	0, 100
Experimental map range	4000 m × 6000 m
Experiment time	3000 s
MAC protocol	IEEE 802.11 p
Radio propagation model	Log Distance Propagation Loss Model
Buffer size	10 MB
Bandwidth	11 Mbps
Transmission range (m)	400, 450, 500, 550, 600
Grid size	1000 m
TTL (s)	10, 20, 30, 40, 50
numPair	50, 100, 150, 200, 250
Beacon time (s)	1
Message time (s)	1, 5, 10, 15, 20

**Table 9 sensors-22-08222-t009:** Performance improvement of SD-QGrid compared to routing protocols with distances.

	Average Delivery Ratio	Average End-to-End Delay	Average Transmission Overhead
QGrid	+17.00%	−27.09%	+1.04%
advQGrid	+10.22%	−31.93%	+1.24%

**Table 10 sensors-22-08222-t010:** Performance improvement of SD-QGrid compared to routing protocols with TTLs.

	Average Delivery Ratio	Average End-to-End Delay	Average Transmission Overhead
QGrid	+21.01%	−24.70%	+1.09%
advQGrid	+13.66%	−34.00%	+1.53%

**Table 11 sensors-22-08222-t011:** Performance improvement of SD-QGrid compared to routing protocols with numPairs.

	Average Delivery Ratio	Average End-to-End Delay	Average Transmission Overhead
QGrid	+13.66%	−27.67%	+0.51%
advQGrid	+9.27%	−35.26%	+1.01%

**Table 12 sensors-22-08222-t012:** Performance improvement of SD-QGrid compared to routing protocols with message time.

	Average Delivery Ratio	Average End-to-End Delay	Average Transmission Overhead
QGrid	+16.47%	−28.65%	+1.42%
advQGrid	+10.77%	−35.08%	+2.31%

## Data Availability

The study did not report any data.

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
