# Peer review of "A Software-Defined Directional Q-Learning Grid-Based Routing Platform and Its Two-Hop Trajectory-Based Routing Algorithm for Vehicular Ad Hoc Networks"

_sensors, 2022, doi:10.3390/s22218222_

Round 1
Reviewer 1 Report
The article is excellent in every aspect. I would suggest including future directions.
Reviewer 2 Report
My research is not exactly in the area of VANET, but more in the area of MANET and wireless sensor networks. Personally, I think that the work is interesting. However, there are several things that I have fund in the text a bit confusing.
The first thing that is unclear (and it is also not clear in the abstract, I think) is the problem that the authors want to solve. The impression is that the authors want to use Q-Elearning and the problem that they want to solve is secondary.
I think that the authors should clearly explain the problem they want to resolve and why it is necessary to do so.
The HELLO protocol is not clear. The text says that it is exchanged between all the vehicles, but if the coverage area is limited, it is only possible to send it to the neighbor's vehicles. Is it propagated by flooding? Or is the hello collected by the RSU and resent to the rest of the nodes?
Due to that, it is not clear how the information is collected, I cannot imagine how the information necessary for the section 3.4.1is obtained. Please, the authors should include some additional information about how the information is collected and shared with the decision module.
In section 3.4.1 step 2 “According to the actual trajectory data, use Formula 3 to calculate the average number of nodes in each grid, and the number of nodes travelling to the adjacent grids in each direction, to generate Tables 3 and 4.” I suppose that is position, for the computation of expression 3 you need the actual position, unless that the algorithm uses the predicted position based in the trajectory, please, clarify this.
Section 3.5, this section is a bit confusing. The authors should explain clearly the necessity of the Two-hop. The explanation is also a bit confusing. In the text the Two hop is for intra grid communication, but the example the protocol can communicate with two hops grids. If is intra grid, I suppose that is to communicate inside the grid, not to communicate to nodes that are until two hop (grids) of distance. Also, as is the redacted, the sensation is that this protocol works similar to an opportunistic routing protocol. Is it true, this? Why is it better to use this method than deliver the packets using the RSU?
Respect to the simulation.
The radio and mac technology should be included, I have supposed that is 802.11p. With the coverage areas used, if you include the buildings obstacles in the radio transmission, with a RSU in the center of the grid many nodes won't obtain the information of the trajectory of the nodes, this can provoke problems in a real use. The sensation that I have with this proposal is that it can be quite vulnerable as the necessary information is lost due to difficulties in the communications.
How many seeds have been used?
It is not a bad idea to include the confidence interval in the figures.
Round 2
Reviewer 2 Report
This version is correct. The only thing I can comment on is, if possible, that the authors can publish the source code in the future. I am in favor, whenever possible, of publishing the source code of the experiments, I know that unfortunately, in many cases it is impossible.